# A multi-hospital, clinician-initiated bacterial genomics programme to investigate treatment failure in severe *Staphylococcus aureus* infections

Stefano G. Giulieri [1,2,3,4] ✉, Marcel Leroi[3,5], Diane Daniel[1,6], Roy Chean[7,8], Katherine Bond[1,2,9], Harry Walker[10,11], Natasha E. Holmes [3,12], Nomvuyo Mothobi[13,14], Adrian Alexander[15], Adam Jenney[15,16], Carolyn Beckett[17], Andrew Mahony[3,18], Kerrie Stevens[6], Norelle L. Sherry [1,3,6] & Benjamin P. Howden [1,3,4,6,9]

Bacterial genomics is increasingly used for infectious diseases surveillance, outbreak detection and prediction of antibiotic resistance. With expanding availability of rapid whole-genome sequencing, bacterial genomics data could become a valuable tool for clinicians managing bacterial infections, driving precision medicine strategies. Here, we present a clinician-driven bacterial genomics framework that applies within-patient evolutionary analysis to identify in real-time microbial genetic changes that have an impact on treatment outcomes of severe *Staphylococcus aureus* infections, a strategy that is increasingly used in cancer genomics. Our approach uses a combination of bacterial genomics and antibiotic susceptibility testing to identify and track bacterial adaptive mutations that underlie microbiologically documented treatment failure (i.e. ongoing positive cultures [persistent infection] or new positive cultures after initial response [recurrent infection]). We show the potential added value of our approach to clinicians and propose a roadmap for the use of bacterial genomics to advance the management of severe bacterial infections.

Thanks to progress in high-throughput whole-genome sequencing (WGS), bacterial genomics has transformed public health microbiology and hospital infection control[1], and is increasingly used to predict antibiotic resistance[2]. However, compared to human and cancer genomics, bacterial genomics has rarely found its application in the clinical setting. This is partly due to turn-around-times that are not suitable for acute infections, but also due to still insufficient evidence regarding the impact of bacterial genetic factors on clinical outcomes[3].

However, the high degree of resolution achieved through bacterial WGS enables an accurate characterisation of the patient's individual bacterial strains, which could inform precision infectious disease management. One of the most promising insights delivered by bacterial genomics is an understanding of the unique evolutionary trajectory of infecting strains during clinical infection. This is particularly the case in microbiologically documented treatment failure, when bacterial cultures remain positive despite antibiotic treatment (persistent infection) or turn positive after an initial response (recurrent infection). Adaptive evolution plays a key role in this setting, driving antibiotic resistance, infection persistence and immune evasion[4], yet it is not routinely assessed in clinical practice. This is in contrast to cancer management, where genomics is used to detect mutations acquired de novo by cancer cells either at diagnosis or during treatment[5]. In cancer, detecting specific mutations can inform individualised management decisions with better outcomes than selecting treatment based on tumour type and

extension[6]. Similarly, the detection of pathoadaptive mutations can shift our interpretation of antibiotic failure in clinical infections[7]. For example, β-lactam resistance in *Staphylococcus aureus* is typically associated with the presence of a modified penicillin-binding protein (PBP2a). In the absence of PBP2a, oxacillin resistance was thought to be due to hyper-production of β-lactamase or PBP mutations. By assessing adaptive evolution in clinical infections, it was possible to show that pathoadaptive mutations drive oxacillin resistance[8]. This has potential clinical implications, as it can help identify oxacillin treatment failure early and inform potential treatment alternatives[9]. Within-host evolution investigations of antibiotic failure have also revealed adaptive mutations promoting antibiotic resistance in persistent infections due to *Mycobacterium tuberculosis*[10,11], *Pseudomonas aeruginosa*[12,13], *Klebsiella pneumoniae*[14], and *Acinetobacter baumannii*[15,16]. Even when no adaptive mutations are detected, comparative genomic analysis of paired baseline and persistent or recurrent isolates can clarify whether apparent treatment failure is due to the ongoing presence of the same bacterial strain or represents reinfection with a genetically distinct lineage. Here, the resolution power of genomics can be leveraged in the same way it is used in outbreak investigations, as shown in genomic studies of recurrent *S. aureus* bacteraemia[17] or *Streptococcus dysgalactiae* prosthetic-joint infection[18].

Here, we propose a bacterial genomics framework to assess adaptive evolution during persistent or recurrent infections. We focus on invasive *S. aureus* infections, due to the clinical relevance of persistence[19,20] and recurrence[17]. We hypothesise that bacterial genomics (supported by phenotypic testing) can assist clinicians in determining the cause of treatment failure in *S. aureus* infections, and potentially guide salvage treatments[20].

## Results

### Description of the analysis framework

Our conceptual framework is outlined in Fig. 1 and Table 1. First, an accurate determination of the genetic distance between isolates collected at baseline and at treatment failure can help ascertain whether persistent or recurrent infections are caused by the same strain or a genetically distinct strain, which requires different management strategies. Depending on their timing, these infections with genetically distinct strains are called superinfections or reinfections (Supplementary Fig. S1). Second, genomics and specialised antibiotic susceptibility testing can reveal previously unrecognised resistance mechanisms. Third, meticulous within-host evolution analysis (both phenotypic and molecular) can identify signatures of adaptive evolution, particularly to antibiotics, information that could be useful when selecting a salvage regimen. Finally, if the above investigations remain negative, this suggests a lack of adaptive evolution. This finding supports continuing the same antibiotic regimen but suggests the presence of a persistent focus, which might warrant more aggressive source control or an increase in the antibiotic dose.

From May 2019 to August 2023, we received 60 clinical strains from 11 episodes of invasive *S. aureus* infections from six hospitals (median two strains per episode, interquartile range 2–4). Details of each episode are provided in Table 2. In all cases, the clinical syndrome was *S. aureus* bacteraemia, of which eight were recurrent, three persistent, and one was both persistent and recurrent. The investigation was initiated by the treating clinician in eight cases and by the clinical microbiologist in three. Antimicrobial susceptibility testing (broth microdilution) showed an increase in oxacillin MIC (≥4-fold) between the baseline and subsequent isolates in two cases, high baseline oxacillin MIC (1 mg/l) in three cases. Cefazolin effect testing was done in three cases and was found to be present in one.

### Within-host evolution analysis of antibiotic failure

Basic genomic data and within-host evolution analysis are shown in Fig. 1 and Table 2. Using a structured interpretation, we categorised antibiotic

failure as likely due to persistent focus in 7 cases and bacterial adaptation in four cases. No case of superinfection/reinfection was found. Phylogenetic analysis of the study isolates and 738 Australian bacteraemia isolates showed that same-patient isolates were monophyletic (Fig. 1B). To confirm the genetic relatedness of the isolates from the same patient, we performed a core-genome multilocus sequence type (cgMLST) analysis. The median pairwise cgMLST allelic distance within the same host, between external isolates and between study isolates and external isolates was four (IQR, 2–8), 1463 (IQR 1377–1646) and 1475 (1379–1653), respectively (Supplementary Fig. S2). Pairwise core-genome SNP distances were similarly segregated between groups, confirming that cgMLST and SNP distance approaches yield comparable results, as previously observed[21]. Thus, our approach provided bacterial genomic evidence of within-host adaptive evolution in a third of sequentially collected strains (Fig. 2A), with adaptive genes including previously described hotspots of adaptation to antibiotic and immune response pressure, like *agrA*[22] (one episode) and *rpoB*[23] (two episodes). This highlights the clinical relevance of bacterial adaptation during invasive *S. aureus* infections and the interest in tracing it using a strategy of sequencing serially collected strains.

Adaptive mutations were observed in genes that have only been recently linked to adaptive antibiotic resistance. For example, mutations in the cyclic-di-AMP phosphodiesterase *gdpP* were found in two of these cases. These were recently shown by multiple groups to drive non-*mec* mediated oxacillin resistance in *S. aureus* (*mec*-independent oxacillin non-susceptible *S. aureus*, MIONSA)[8,9,24–26]. Mutations in the regulatory serine-threonine phosphatase *stp1* were detected in two isolates. This gene, which has been linked to vancomycin[27] and oxacillin resistance[28], displayed one of the strongest statistical signals of adaptive evolution in our large-scale within-host evolution analysis of almost 400 episodes of *S. aureus* infections[4].

In addition to ranking mutations based on the significance of the within-host mutations enrichment, we reconstructed within-host phylogenies in selected episodes with multiple isolates to track the evolutionary trajectory of each individual infection episode[29] (Fig. 3B, C). In one episode with multiple adaptive mutations, we observed two independent acquisitions of *tagO* mutations (Fig. 3B). This suggested within-host convergent evolution, a powerful marker of adaptive evolution[30] that could be used as an additional evolutionary genomics tool to investigate antibiotic treatment failure when multiple isolates are available.

### Clinical impact of the within-host evolution analysis report

To assess the clinical impact of our within-host evolution analysis approach, we created a survey containing a clinical vignette and the within-host evolution analysis report of the 11 cases (Supplementary Table S3). The survey was filled out by a panel of 25 infectious diseases physicians based in Australia, Switzerland, the United Kingdom, and the United States, of which 12 were *S. aureus* researchers, 9 were clinical microbiologists, and 7 had expertise in bacterial genomics. Clinicians scored report readability and usefulness at 80 out of 100 (IQR 60–90). Since a key piece of information in our report was the presence/absence of adaptive mutations, we sought to investigate how good clinicians are at inferring their presence without genomic data (i.e., based on clinical and routine microbiological information). We reasoned that the added value of our within-host evolution analysis would be maximal if clinicians were not able to predict adaptive mutations with high accuracy. Overall estimated median probability of adaptive mutations was 30 (IQR 20–70). Adaptive mutations were correctly predicted by >50% of the respondents only for two out of four episodes (Fig. 3). We observed that providing the within-host evolution report led to a change in the antibiotic plan in 34% of responses ($p = 0.01$, Chi-squared test). In a multivariate logistic model, providing a genomic report and the presence of adaptive mutations were both independently associated with the decision to recommend a

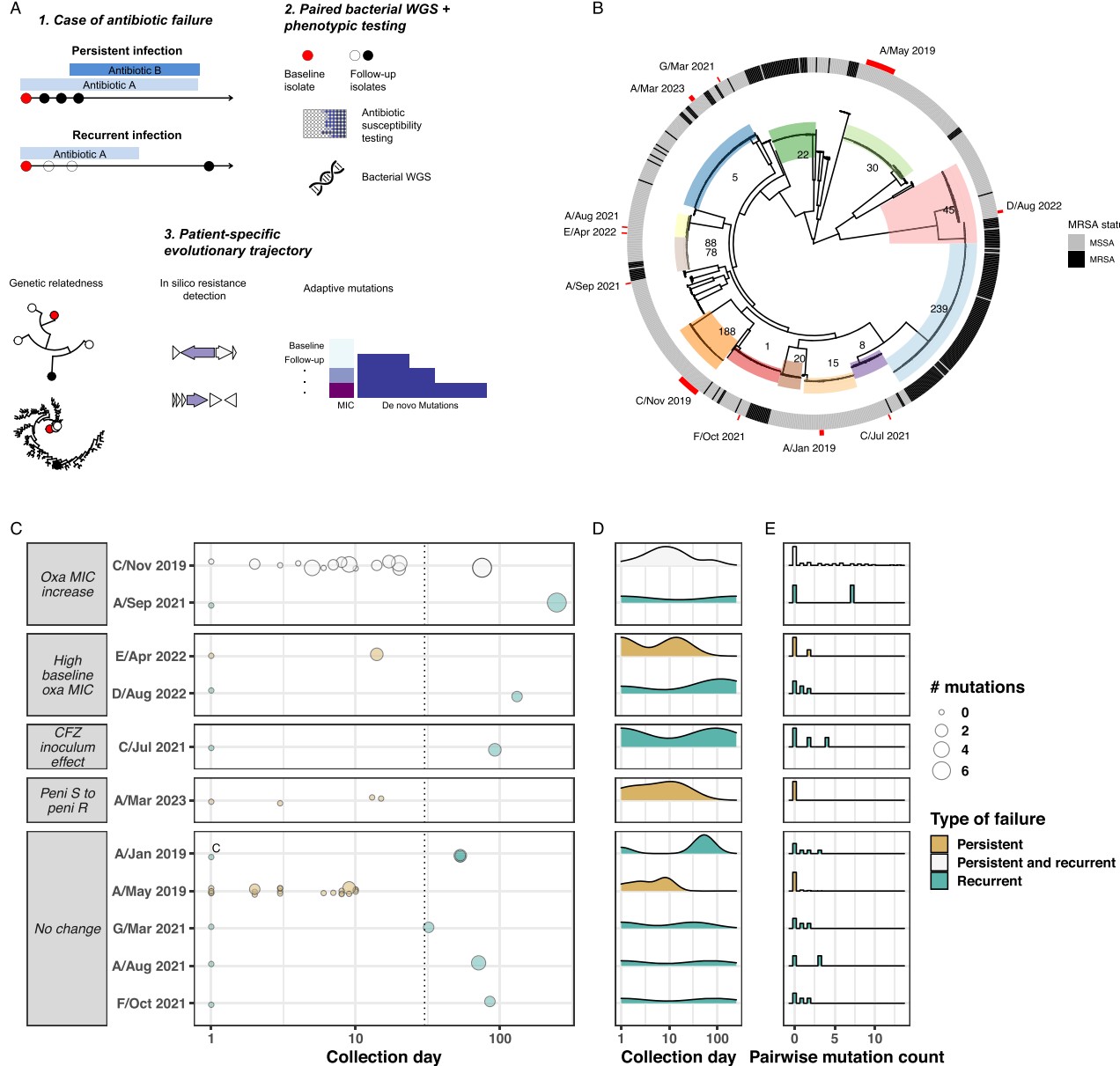

**Fig. 1 | Overview of study approach and results. A** Proposed approach to identify and track bacterial adaptive mutations underlying antibiotic treatment failure. WGS: whole-genome sequencing. **B** Maximum-likelihood phylogenetic tree of 798 *S. aureus* bacteraemia strains showing the genetic background of isolates from antibiotic treatment failure. Of note, all same-patient isolates belong to monophyletic clades. MRSA: methicillin-resistant *S. aureus*. MSSA: methicillin-susceptible *S. aureus*. **C** Timeline, observed phenotypic changes, and number of within-host mutations (relative to a baseline isolate). Oxa oxacillin, MIC minimum inhibitory concentration, CFZ cefazolin, S susceptible, R resistant. **D** distribution of isolate collection days (after baseline isolate) between persistent and recurrent case. **E** Distribution of within-host mutation pairwise counts. Source data are provided as a Source Data file.

## Table 1 | Analysis framework

| Clinical question | Bacterial genomics information | Potential clinical impact |
|---|---|---|
| Reinfection with new strain/clone? | Large genetic distance between baseline and follow-up strain | Look for source of reinfection—no change of treatment |
| Antibiotic resistance not detected by routine testing | Detection of resistance genes/mutations in both baseline and follow-up strain | Change antibiotic treatment |
| Antibiotic adaptation | Detection of 'driver mutations' in the follow-up strain | Consider change antibiotic substance or class (consider cross-resistance) |
| Persistent infection focus? | Minimal genetic distance between baseline and follow-up strain | Aggressive focus search, source control, and increase antibiotic dose |

**Table 2 | Description of cases**

| Hospital | Month/year | aIsolates | Clinical syndrome | Persistent/recurrent | Initiated by | Phenotypic finding | Sequence type | Resistance genes | aMutations | Final assessment | Genes with adaptive mutations |
|---|---|---|---|---|---|---|---|---|---|---|---|
| A | | 4 | SAB | Persistent | Clinician | Baseline: penicillin S. Follow-up: penicillin R | novel | | 0 | Persistent focus | |
| A | | 2 | SAB | Recurrent | Microbiologist | Oxacillin MIC increase | 25 | blaZ | 7 | Adaptation | stp1, gdpP, fmtA |
| A | | 2 | SAB | Recurrent | Clinician | None | novel, 88 | blaZ | 3 | Persistent focus | |
| C | | 16 | SAB | Persistent, Recurrent | Microbiologist | Oxacillin MIC increase | 188 | blaZ | 24 | Adaptation | stp1, gdpP, rpoB, pbp1, pbp3, pbp4 |
| D | | 3 | SAB | Recurrent | Clinician | High baseline oxacillin MIC | novel | blaZ | 1 | Persistent focus | |
| C | | 2 | SAB | Recurrent | Clinician | Cefazolin inoculum effect | 8 | blaZ | 2 | Adaptation | parC |
| E | | 2 | SAB | Persistent | Microbiologist | High baseline oxacillin MIC | 88 | blaZ | 2 | Persistent focus | |
| F | | 2 | SAB | Recurrent | Clinician | None | novel | blaZ, dfrG, fusC | 1 | Persistent focus | |
| G | | 2 | SAB | Recurrent | Clinician | None | 5 | blaZ | 2 | Persistent focus | |
| A | | 4 | SAB | Recurrent | Clinician | None | 15 | blaZ | 4 | Adaptation | rpoB |
| A | | 21 | SAB | Persistent | Clinician | None | 30 | blaZ | 34 | Persistent focus | |

aMutations: total number of point mutations identified when comparing subsequently collected strains to the baseline strain. SAB Staphylococcus aureus bacteraemia, MRSA methicillin-resistant S. aureus, MIC minimum inhibitory concentration.

change in antibiotic regimen. While the report led to changes in antibiotic duration (17%), and in decisions regarding source control (13%), these were not significant (Supplementary Tables S1 and S2). Overall, the survey suggests that supplementary phenotypic and genomic information provides added value to the assessment of antibiotic failure and could inform decisions on antibiotic therapy. We also show that adaptive mutations can't always be predicted based on the clinical history and routine microbiological results.

## Discussion

Treatment failure in *S. aureus* invasive infections is challenging for both patients and clinicians. Our clinician-initiated, genomics-informed approach highlights the potential contribution of within-host evolution analysis to its investigation. We show that an evolutionary genomics framework can provide useful answers for clinical management, for example, by identifying mutations in genes that are associated with pathoadaptation or by providing molecular evidence of infection relapse from a persistent focus.

To assess the potential clinical impact of our phenotypic and genomic within-host evolution analysis, we provided a report to an international group of infectious diseases physicians and recorded their management decisions before and after reading the report. Our survey showed the added value of additional phenotypic and genomic investigations and suggested that the awareness of these results can inform treatment decisions in up to 30% of cases. It also indicated that it is challenging to predict whether adaptive mutations are present based solely on clinical presentation and routine microbiological results. Overall, this suggests that within-host evolution analysis might provide valuable information to clinicians managing antibiotic treatment failure. This approach should be investigated prospectively in a larger number of cases. Crucially, patient outcomes should also be assessed, as changes of practice or subjective added value don't necessarily correlate with improved clinical outcomes.

While the clinical impact of bacterial genomics has been demonstrated for infection control[31] and antibiotic resistance prediction (typically at the start of treatment)[32], the role of genomics in informing management of treatment failure has yet to be explored. Within-host evolution studies have leveraged sequential samples to discover mechanisms of antibiotic resistance or understand the evolutionary dynamics of emergent resistance[33]. However, the potential usefulness and gain of information of performing a simple paired WGS is often not considered when discussing the investigation and management of persistent bacterial infections[34]. Here, we provide preliminary data indicating that within-host evolution phenotypic and genomics analyses deliver additional information that cannot be easily inferred from the clinical context and routine microbiological data (genetic relatedness, adaptive mutations), and this information may lead to changes in the salvage treatment. Importantly, until the full spectrum of adaptive mutations driving antibiotic resistance and treatment failure is revealed, parallel phenotypic testing is necessary to avoid missing the emergence of resistance that is not yet predicted by the detection of known de novo mutations.

When assessing the utility of genomic investigations of clinical bacterial isolates, two key potential barriers to implementation are the additional costs of sequencing and whether results can be delivered in a clinically meaningful timeframe. While we acknowledge that costs will vary significantly depending on location, existing access, infrastructure, sequencing platforms and workflows, and arrangements with suppliers, we estimated the cost of a within-host evolution analysis of two clinical isolates in our laboratory (high-throughput genomic sequencing) to be approximately $AUD 584 (Australian dollars, ~ $US 370). This is in line with a recent systematic review, which calculated that the mean cost of bacterial WGS was US $194[35]. Equally crucial is the turnaround time of the analysis, which in our laboratory is 5–7 days at a minimum. Shorter turn-around times can be achieved by

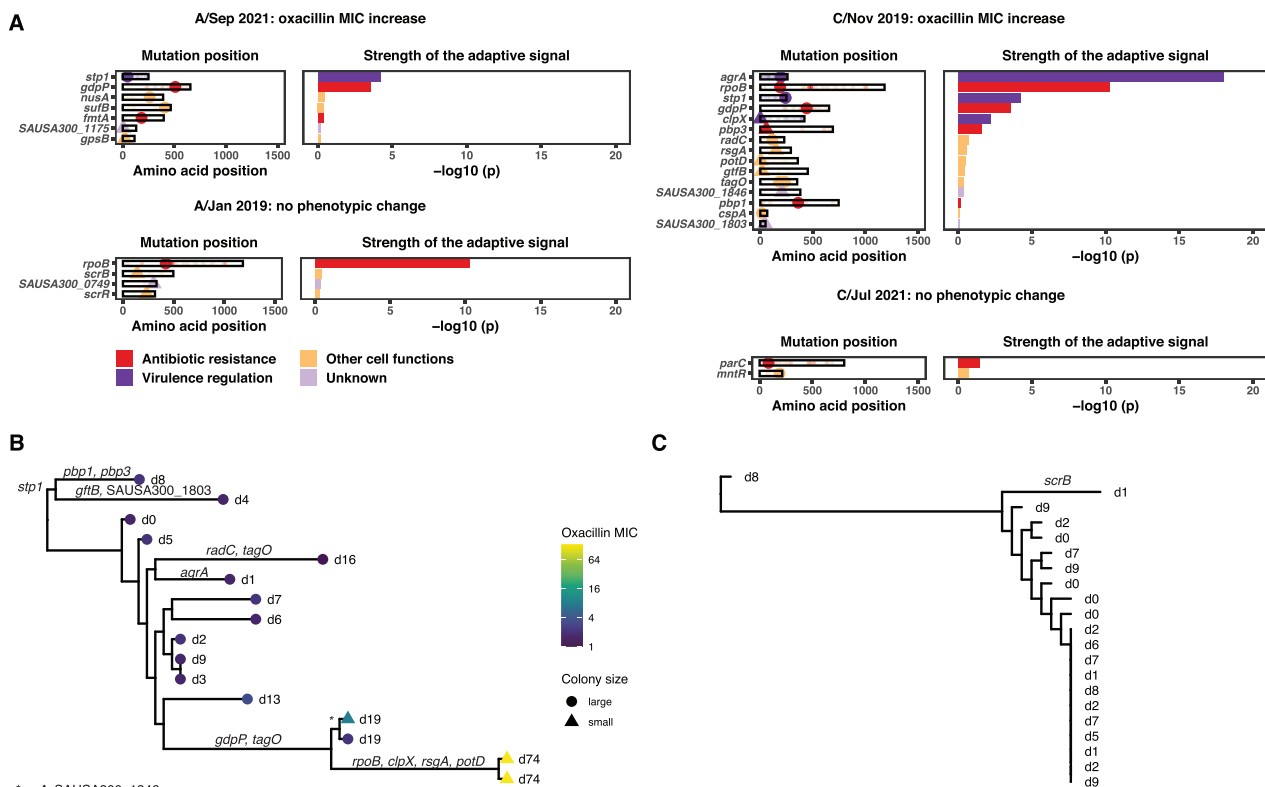

**Fig. 2 | Within-host evolution analysis of adaptive episodes. A** Detailed within-host evolution analysis of four cases with evidence of adaptive mutations. The first panel shows the location of the identified mutation. The second panel shows the statistical significance of the enrichment (adaptive signal: Poisson regression, one-sided likelihood ratio test, no adjustment for multiple comparisons) in a large-scale within-host evolution analysis of 400 *S. aureus* infection episodes. Dot and bars are coloured according to the functional association of the mutated gene. MIC minimum inhibitory concentration. **B, C** Reconstruction of within-host evolutionary trajectories of two cases with >2 clinical isolates. The maximum-likelihood trees are inferred from a core-genome alignment on the internal reference. Trees were rooted using the closest available isolate from the bacteraemia collection used for the global phylogeny. Tips are annotated with the day of collection, with the baseline isolate displayed as the day 0 isolate. Inferred emergence of non-synonymous substitutions and truncations is annotated on the internal nodes or branches. **B** Within-host phylogenetic tree of Case C/Nov 2019 persistent and relapsing bacteraemia with progressive oxacillin MIC increase and emergence of small colony variants. **C** Within-host phylogenetic tree of Case A/May 2019: no phenotypic changes. Source data are provided as a Source Data file.

on-demand sequencing, however the per-sample cost for this workflow will be higher than for high-throughput sequencing workflows[36]. The location of sequencing also contributes to turnaround time, with local sequencing (e.g., at the hospital laboratory) likely to have lower turnaround times than sequencing at a centralised referral laboratory. The critical questions around implementation should be addressed in a larger, prospective study, where a standardised report is provided to clinicians managing treatment failure. The clinical impact of the reported results (antibiotic choice, antibiotic duration, source control recommendation) can be assessed using the survey approach that we propose in this study, with participation of both the involved clinicians and external assessors not involved in the case. Beyond clinical impact, turnaround times and health economic aspects should also be assessed. For example, the cost-effectiveness of the genomic investigation could be increased by a more targeted selection of cases (i.e., strict definitions of persistent and recurrent infection) or by using cheaper non-genomic microbiological tests as initial screening (e.g. MALDI-TOF to infer genetic relatedness, currently in development).

Given that turnaround time is a limiting factor of bacterial genomic analyses of clinical isolates, a potentially attractive option is using long-read Oxford Nanopore sequencing. Studies have shown that this technology can detect antibiotic resistance genes within hours of sample collection[37]. In addition, long reads allow to detect chromosome structural variants more effectively, which can be relevant in the setting of persistent *S. aureus* infections[38]. While the accuracy of variant calling from long reads is improving[39], it is still reliant on bespoke

pipelines that might not be routinely available. Given the lack of an established analysis approach for questions including within-host evolution and the increased costs due to the manual workflow, we believe that short-read sequencing remains the preferred choice for antibiotic treatment failure analysis. However, future studies should compare the relative accuracy, costs and turnaround times of short and long-read sequencing for this type of analysis.

While the work presented here is limited to a small number of episodes and to a single bacterial species, it provides the proof-of-concept for a structured approach to identify and track bacterial adaptive mutations underlying antibiotic treatment failure using a combination of bacterial genomics and antibiotic susceptibility testing. Full assessment of clinical impact and utility was not included in the current study, but importantly, needs to be addressed in future work.

## Methods
### Ethics approval
Human research committee approval was obtained at the Austin Health Human Research Ethics Committee (HREC/105787/Austin-2024), Melbourne, Australia, which provided a waiver of consent for this retrospective case series.

### Identification of cases of antibiotic failure
In a preliminary study, we offered our genomic investigation framework to teaching hospitals in the state of Victoria, Australia, between 2020 and 2023. The investigations were initiated by clinicians in cases

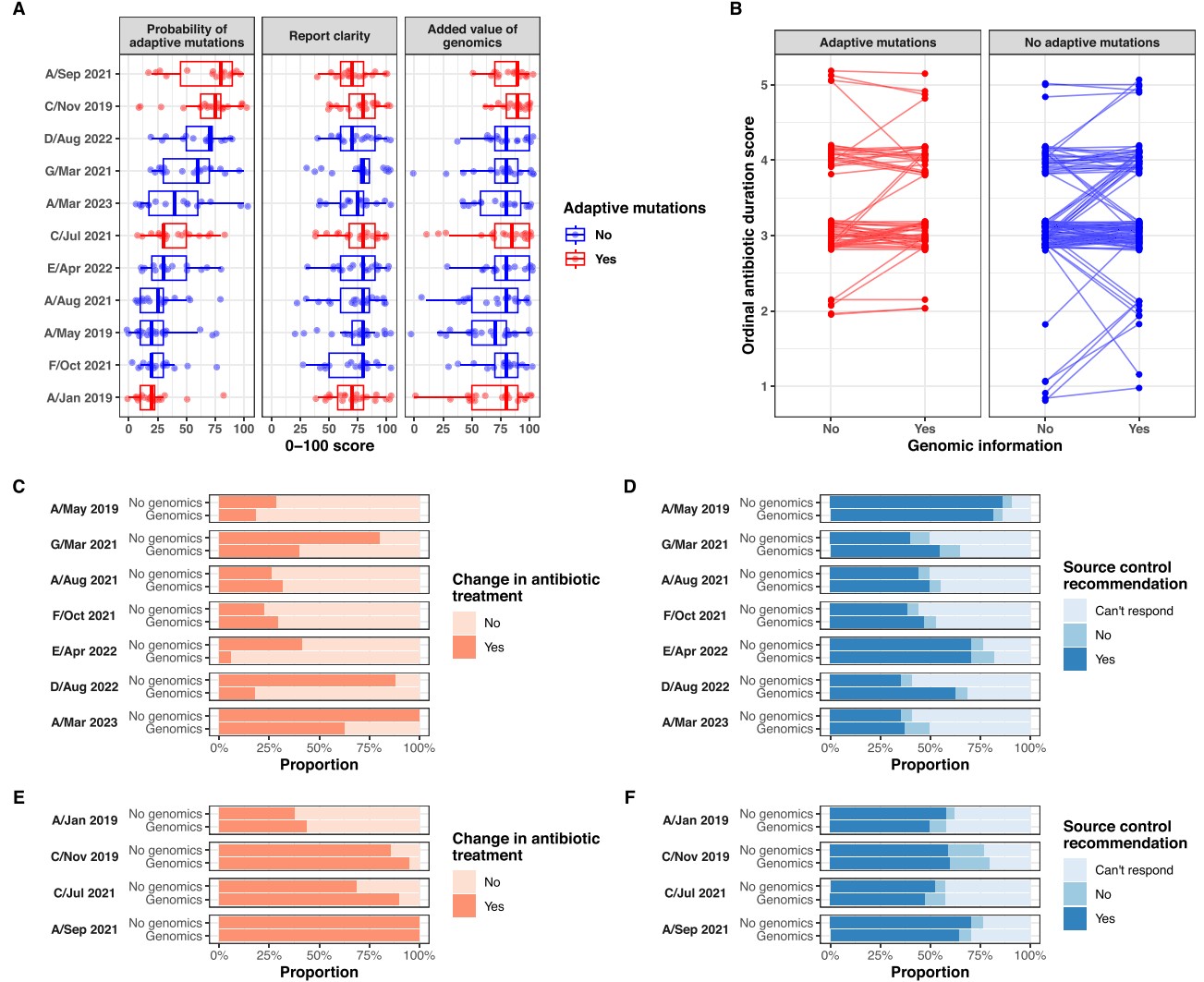

**Fig. 3 | Overview of the RedCap survey of 25 infectious diseases specialists.**
**A** Clinicians assessment of probability of adaptive mutations, report readability, and added value, using a score between 0 and 100. The box plots represent median, interquartile range (hinges), and 1.5 times the interquartile range (whiskers), with outliers represented as individual dots. **B** Changes in recommended antibiotic duration (based on an ordinal score) after providing additional phenotypic and genomic information. **C**–**E** antibiotic switch (**C**, **E**) and source control (**D**, **F**) recommendations before and after reading additional phenotypic and genomic information, for cases with (**E**, **F**) and without adaptive mutations (**C**, **D**). Source data are provided as a Source Data file.

of suspected antibiotic treatment failure in invasive *S. aureus* infections indicated by microbiological persistent (ongoing positive cultures after ≥5 days) or recurrent infection (new positive cultures after initial response), or unusual phenotypic characteristics of clinical strains noted by medical microbiologists (e.g. increase in minimum inhibitory concentration [MIC], small colony variants). Clinical data collected included *S. aureus* infection syndrome, duration of bacteraemia, timing and symptoms of recurrence, and infection treatment. No demographic and comorbidities data were collected.

**Phenotypic testing**
Strains were referred to the Microbiological Diagnostic Unit, Public Health Laboratory in Melbourne, Australia, where they underwent broth microdilution antibiotic susceptibility testing using the Sensititre® GPN3F and GN6F panels and EUCAST interpretive breakpoints. In addition, the cefazolin inoculum effect was assessed as described in ref. 40. Briefly, cefazolin MICs were determined using broth microdilution with a standard ($5 \times 10^5$ cfu/ml) and high inoculum ($5 \times 10^7$ cfu/ml) dilution. *S. aureus* ATCC 25923 was used as the negative control strain. A > 4-fold increase in the cefazolin MIC with the high inoculum compared to the MIC obtained with the standard inoculum was considered evidence of the cefazolin inoculum effect.

**Bacterial WGS**
For bacterial WGS of the same-episode strains, single colonies were isolated from the referred sub-cultures and suspended in lysis buffer. Bacterial DNA was extracted on the QIAsymphony using the DSP Virus/Pathogen Mini Kit. Library preparation was carried out using Nextera XT (Illumina Inc.). WGS was performed on the NextSeq 500/550 platform (Illumina Inc.), producing 150 bp paired-end reads. Quality control and reads assembly were performed using a standardised pipeline (https://github.com/MDU-PHL/mdu-tools/blob/master/bin/mdu-qc) that calculates reads depth and quality, computes the fraction of *S. aureus* reads using Kraken v2.1.1[41] and assembles reads using Shovill v1.0.1 (https://github.com/tseemann/shovill), based on SPAdes v3.14.1[42]. Assemblies were annotated with Prokka v1.14.6[43]. Multilocus sequence type (MLST) was inferred from the assembled contigs using the Mlst tool v2.19.0 (https://github.com/tseemann/mlst), and resistance genes were detected with Abricate v1.0.1 (https://github.com/tseemann/abricate) using the NCBI AMRFinderPlus database[44].

## Phylogenetic analysis

To put clinical cases of antibiotic failure in the broader context of the *S. aureus* population structure, we combined clinical case isolates reads with a previously published collection of 738 Australian *S. aureus* bacteraemia genomes[45]. Reads were mapped to the reference genome *S. aureus* BPH2947 (accession GCF_900620245.1)[46] using Snippy v4.6.0 (https://github.com/tseemann/snippy). A core alignment including all positions where ≥90% genomes had ≥10 coverage depth[47] was built using Snippy-core v4.6.0, Goalign v0.3.4[48] and SNP-sites v2.5.1[49]. A maximum-likelihood phylogenetic tree was inferred using IQ-TREE, v2.0.3[50], under a GTR + G4 model. The phylogenetic tree and related metadata were visualised in R using the ggtree package, v3.12.0[51]. Pairwise SNP distances were calculated using SNP-dists, v0.7.0 (https://github.com/tseemann/snp-dists).

## Core-genome multilocus sequence type analysis

We used chewBBACA, v3.3.10[52], to generate a core-genome multilocus sequence type (cgMLST) profile for the 798 isolates included in the global phylogeny. We adapted the publicly available *S. aureus* cgMLST schema (downloaded from https://www.cgmlst.org/ncs/schema/Saureus1244) using the command "chewBBACA.py PrepExternalSchema". We used the prepared schema to call alleles from the fasta nucleotide files with the coding DNA sequences (CDS) of the assemblies (command "chewB-BACA.py AlleleCall") and kept genes present in 95% of the assemblies (command "chewBBACA.py ExtractCgMLST"). We computed a pairwise distance matrix using cgmlst-dists v0.4.0 (https://github.com/tseemann/cgmlst-dists).

## Within-host evolution genomic analysis

The within-host evolution genomic analysis (i.e., comparative genomics of bacterial isolates collected at the time of failure vs. baseline isolates) was performed using a bespoke pipeline as described in ref. 4 and available at ref. 53. This pipeline uses Snippy to call variants using the baseline strain draft assembly as a reference. To increase the accuracy of the variant calling, additional filtering steps are added as described in ref. 38. Within-host genetic relatedness was defined as less than 100 mutations[4]. To infer adaptive mutations, we used our previously published curated collection of serially sequenced episodes of invasive *S. aureus* infections[4], supplemented with three studies that were published since the last analysis[8,17,54]. We generated Poisson models of gene-specific mutation counts for each gene with a homologue in reference genome *S. aureus* FPR3757 (NCBI accession CP000255) and inferred the statistical significance of the within-host mutation enrichment by comparing gene-specific models with a neutral evolution model using likelihood ratio[4,55]. We searched the list of mutated FPR3757 homologues on the platform Microbesonline (https://microbesonline.org/) to extract the functional annotation of the mutated genes and to infer the main functional category. In addition, for antibiotic resistance, we performed a literature search using the gene name and the terms "*Staphylococcus aureus*" and "antibiotic resistance". We selected one functional category per gene. In cases of multiple functional categories, we prioritised "Antibiotic resistance", followed by "Virulence regulation".

## Within-host phylogenies

To infer within-host phylogenies, we generated a core-genome alignment from reads mapped to the internal reference (i.e., draft assembly of the earliest available invasive isolate). The alignment included all isolates from the infection episode and the closest external isolate included in the global phylogeny (that is, among 738 previously published Australian *S. aureus* blood isolates[45]). To identify the closest external isolate, we extracted the most recent common ancestor of the episode isolates, obtained its parent node and selected the descendant that didn't belong to the episode. A maximum-likelihood phylogenetic tree was inferred as described above. The external isolate was used a outgroup to root the tree.

## Development of a clinical reporting approach

Clinicians were provided with two reports: the first included standard antimicrobial susceptibility and the cefazolin inoculum effect results, plus basic genomic characterisation (MLST and resistance gene detection). The second report described the within-host evolutionary analysis and proposed an interpretation based on the structured approach described above (Table 1) and available published evidence regarding the identified mutations.

## Assessment of clinical acceptability of reports

To assess the acceptability and impact of the analysis, we developed a RedCap survey with a description of each case with and without phenotypic and genomics information. Infectious disease physicians not involved in the management of the cases were invited to participate in the survey. The survey included unique questions about the readability and added value of the report and the clinician's assessment of the probability of the presence of adaptive mutations. For these questions, clinicians were asked to select a score between 0 and 100 with increments of 10. In addition, we asked clinicians to indicate before and after reading the report: (i) interpretation of antibiotic failure, (ii) request for additional investigations; (iii) proposed duration of antibiotic treatment (using an ordinal score ranging from 1 (≤2 weeks) to 5 (>12 weeks); (iv) proposed antibiotic class switch; (v) source control recommendation.

## Statistical analysis

To assess whether our genomic report had an impact on the treatment recommendations made by clinicians participating to the RedCap survey, we excluded respondents with >50% missing answers and assessed the association between treatment recommendations and potential predictors (genomic report, presence of adaptive mutations, country of residence and professional background of the respondents) by logistic regression or linear regression analysis. We performed a univariate and multivariable regression using the same variables (i.e., no reduction of variables in the multivariable model). All analyses were carried out in R, version 4.4.1.

## Reporting summary

Further information on research design is available in the Nature Portfolio Reporting Summary linked to this article.

## Data availability

Sequence data that support the findings of this study have been deposited in NCBI under accession code PRJEB85744. Source data are provided with this paper.

## Code availability

The code and scripts used to generate the results of the within-host evolution analysis are publicly available in Zenodo (https://doi.org/10.5281/zenodo.15258746) and Github (https://github.com/stefanogg/staph_adaptation_paper).

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

## Acknowledgements

This work was supported by a Research Fellowship from the National Health and Medical Research Council, Australia to B.P.H. (GNT1196103) and by a Melbourne Genomics Immersion Fellowship to S.G.G. We acknowledge all the clinicians who answered the RedCap survey: Eugene Athan (Barwon Health, Geelong, Australia), Enos Bernasconi (Ente Ospedaliero Cantonale, Lugano, Switzerland), Silvio Brugger (Zurich University Hospital, Zurich, Switzerland), Justin Denholm (Royal Melbourne Hospital, Melbourne, Australia), Benoit Guery (Lausanne University Hospital, Lausanne, Switzerland), Oriol Manuel (Lausanne University Hospital, Lausanne, Switzerland), Matthaios Papadimitriou-Olivgeris (Lausanne University Hospital, Lausanne, Switzerland), Joshua Parsons (Duke University Hospital, Durham, United States), Trisha Peel (Alfred Health, Melbourne, Australia), Neta Petersiel (Monash Medical Centre, Melbourne, Australia), Owen Robinson (Royal Perth Hospital, Perth, Australia), Jacques Schrenzel (Geneva University Hospitals, Geneva, Switzerland), Steven Tong (Royal Melbourne Hospital, Melbourne, Australia), Sebastiaan Van Hal (Royal Prince Alfred Hospital, Sydney, Australia), Bernadette Young (Nuffield Department of Medicine, Oxford, United Kingdom). We thank Susan Ballard (Microbiological Diagnostic Unit, Public Health Laboratory, Melbourne, Australia) for key insights and for providing data on sequencing costs and turnaround times. We thank Melissa Martyn (Melbourne Genomics Health Alliance) for reviewing the RedCap survey. We thank Jason K. Kwong (Austin Health, Melbourne, Australia) for critical review of the study protocol and study support.

## Author contributions

Conceptualisation, S.G.G., N.L.S., and B.P.H.; methodology, S.G.G., D.D., K.S., and N.L.S.; software, formal analysis, investigation, S.G.G., M.L., R.C., K.B., H.W., N.E.H., N.M., A.A., A.J., C.B., and A.M.; data curation, S.G.G. and D.D., writing—original draft: S.G.G.; writing—review & editing: S.G.G., M.L., K.B., N.L.S., B.P.H.; visualisation: S.G.G.; supervision: N.L.S. and B.P.H.; project administration: S.G.G.; funding acquisition: B.P.H.

## Competing interests

The authors declare no competing interests.

## Additional information

[1]Department of Microbiology and Immunology, The University of Melbourne at the Doherty Institute for Infection and Immunity, Melbourne, VIC, Australia. [2]Victorian Infectious Disease Service, The Royal Melbourne Hospital, at the Peter Doherty Institute for Infection and Immunity, Melbourne, VIC, Australia. [3]Department of Infectious Diseases and Immunology, Austin Health, Heidelberg, VIC, Australia. [4]Centre for Pathogen Genomics, The University of Melbourne, Melbourne, VIC, Australia. [5]Department of Pathology, Austin Health, Heidelberg, VIC, Australia. [6]Microbiological Diagnostic Unit Public Health Laboratory, The University of Melbourne at the Doherty Institute for Infection and Immunity, Melbourne, VIC, Australia. [7]Department of Microbiology, Eastern Health, Box Hill, VIC, Australia. [8]Department of Infectious Diseases, Eastern Health, Box Hill, VIC, Australia. [9]Department of Microbiology, Royal Melbourne Hospital, Parkville, VIC, Australia. [10]Dorevitch Pathology, Footscray, VIC, Australia. [11]Department of Infectious Diseases, Western Health, Footscray, VIC, Australia. [12]Department of Infectious Diseases, The University of Melbourne at the Peter Doherty Institute for Infection and Immunity, Melbourne, VIC, Australia. [13]Department of Infectious Diseases, University Hospital Geelong, Geelong, VIC, Australia. [14]AMR-One Health Working Group, Burnet Institute, Melbourne, VIC, Australia. [15]Department of Infectious Diseases, The Alfred Hospital, Melbourne, VIC, Australia. [16]Microbiology Unit, The Alfred Hospital, Melbourne, VIC, Australia. [17]Epworth HealthCare, Richmond, VIC, Australia. [18]Department of Infectious Diseases, Bendigo Health, Bendigo, VIC, Australia. ✉e-mail: stefano.giulieri@unimelb.edu.au

