## [Transparent Peer Review file · Nature Communications]

A multi-hospital, clinician-initiated bacterial genomics program to investigate treatment failure in severe *Staphylococcus aureus* infections

Corresponding Author: Dr Stefano Giulieri

Version 0:

Reviewer comments:

Reviewer #1

(Remarks to the Author)

Giulieri and colleagues report on the utilization of whole genome sequencing strategy to detect treatment failure in patients suffering severe *Staphylococcus aureus* infection. The authors used comparative genomics techniques to detect emergence/selection of mutations triggering antibiotic resistance in order to adapt optimally antimicrobial treatment. Monitoring the optimal treatment in real-time is challenging but will represent an inestimable benefit for the patient.

General remarks: The paper is short, clear and the results are definitely informative. The field of research is particularly relevant and the work conducted by a renowned group, very active in the field. The work appears rigorous and the manuscript is well-structured. The conclusions are in accordance with the results obtained by the authors. In conclusion, I see no mandatory changes to require and I would only suggest minor changes or comments.

- The manuscript contains a couple of typos (check l. 125; *S. aureus* not italicized l. 236; always l. 161).
- The study requires real-time sequencing; I would suggest the authors to introduce a (short) section about cost of the analysis.
- The idea of contacting a large panel of ID specialists is really relevant
- My main comment relies on the information obtained by WGS. I would suggest to add a paragraph about the limitations of the study. Indeed, some resistances are not linked to emergence/selection of mutations, phenotypical tests appear mandatory. In addition, it appears obvious that the method is mainly adapted to persistent infections but not to relapsing ones; in the latter situation the strain could evolved silently for months or years without the possibility to detect any infectious episode.
- I hope a larger study allowing assessing benefit of the method will be initiated in the future by the authors

Reviewer #2

(Remarks to the Author)

I appreciate that this is a pilot study of 11 cases, and as such, the data are limited. I applaud the authors for clearly and accurately describing the limitations of this study and the the requisite next steps (critically, looking at patient outcomes). While I agree that the technology shows promise, there are several ways in which that case is not strongly made by the manuscript as it is currently presented.

Major:

- 1) The statement that "Overall, the survey suggests that supplementary phenotypic and genomic information provides added value to the assessment of antibiotic failure and could inform decisions on antibiotic therapy." does not seem strongly supported. The authors must present statistical analysis to indicate whether any meaningful shift in the metrics they are

presenting occurred with the presentation of users with genomic data for any of the metrics they are collecting (ie the data in Figure 3, panels B through F).

If these shifts in provider decision making do not pass statistical significance in this small pilot study it is not necessarily a deal-breaker, but this statistical analysis should be performed and presented. If statistical significance is not achieved, are there features of particular cases where genomic data was most informative?

2) Some of the language used is inconsistent with is a bit confusing and needs to be better clarified or defined, in my opinion. Infections are described, alternately, as "persistent" vs "recurrent" vs "re-infection" vs "relapse" vs "adaptive". My understanding (albeit limited) is that "persistence" refers to a state where a small subpopulation of bacteria, called "persisters," remain alive and dormant within the body even after treatment, potentially leading to a relapse of the infection later on, while "recurrence" more generally means the infection comes back again, which could be due to either persistence (same bacteria reactivating from a dormant state) or reinfection with a new strain of the bacteria from an external source. In this framework, recurrence and persistence are not mutually exclusive. This is all a way of saying that the authors should formally define persistence and recurrence in the abstract, and relate these definitions to both the empiric measurements they are making about total and adaptive mutations, and to the other terminology they are using (ie, relapse).

3) Line 154: "Overall probability of adaptive mutations was 30 (interquartile range 20-70)." This statement is unclear to me, and seems to conflict with the following statement that "adaptive mutations were correctly predicted only for two out of four episodes".

4) Figure 3. "Clinicians assessment of probability of adaptive mutations" is being measured. It is unclear why this metric is useful or is being tracked. Shouldn't the empiric data define whether there are adaptive mutations present in the strains being examined? Why and how is this the opinion of the provider receiving the report, instead of a presentation of fact?

5) In the case of emerging antibiotic resistance, adaptive mutation would be of obvious importance, but it is not clear how *S. Aureus* adaptations during chronic disease that modulate immune response pressure (eg, *agrA*) are useful. As it relates to informing clinical decision making, I feel that the authors need to more cleanly focus on adaptive mutations relating to antibiotic resistance rather than those related to other stigmata of chronic disease, or at least to divide such mutations into discrete categories.

6) The form of the analysis report is described, but with only 11 cases contained in this study it would be beneficial and appropriate for the authors to present the actual reports issued for each of the 11 cases (ie, the pair of reports issued to the clinicians) as supplemental information.

7) Genomic data availability: The sequencing data generated from this study should be made publicly available in the SRA or equivalent database.

Minor:

1) Figure 1 legend is incomplete "Maximum likelihood phylogenetic tree of XX *S. aureus* bacteraemia strains".

2) Line 264. The source of the "closest external isolate" used is unclear. Is this a genome from a public repository unrelated to this study, or was some operation performed to identify an isolate from the study cohort that satisfied this requirement? This is alluded to in the figure legend but should be cleanly communicated in the methods.

3) line 176 : "Our survey showed the added value of additional phenotypic and genomic investigations and suggested that the awareness of these results can inform treatment decision in up to 30% of cases." How is that statistic/conclusion being reached? Does this refer to the 34% of responses in changes in antibiotic choice?

Reviewer #3

(Remarks to the Author)

The study by Giuleri et al. aims to show the usefulness and potential of clinical microbial (bacterial) genomics in real-time to identify and track bacterial adaptive mutations related to antibiotic treatment failure. The identification of pathoadaptive mutations driving oxacillin resistance as well as the discrimination between persistence/relapse/reinfection using WGS has been shown. Furthermore a roadmap for its use is proposed based on the interaction with clinicians.

The study is methodologically correct, well structured and aims to support/encourage the implementation of microbial genomics in the clinic aiming for a genomics-informed treatment and not only *a posteriori* analysis. Though most of my questions have been answered in the manuscript, some key aspects might require further discussion/detail to strengthen the arguments and the proposal.

Below I provide some comments/questions that when addressed might help refine the manuscript.

-As one of the factors preventing the implementation of bacterial genomics within the clinical practice is mentioned "turn-around-times that are not suitable for acute infections", however no data is provided regarding a timeline (crucially duration) from sample acquisition to genomic analysis results being provided.

-Considering the increased accuracy provided by the novel ONT flow cells, the added value of having long reads to identify

genetic rearrangements and the known low turn-around-time. Do the authors believe it would be possible to use this alternative technology to perform the same analysis? I believe adding this to the discussion would be valuable. Specially interesting for those having this technology already.

-An additional column in table 2 specifying Month/Year of the episode would help to easily follow the data in the figures.

-Considering that all samples do not belong to the same CC. Why was chosen a SNP analysis over a cgMLST analysis for the ML phylogenetic tree of Figure 1B?

-Considering that there are cases where not additional information was provided by WGS analysis (i.e. phenotype changed and association with genomic data not possible, such as the case in hospital A, Mar 2023) and that the costs of performing WGS are still considerable. Is the data presented enough for a strong argument supporting the implementation of WGS to support treatment decisions? Discussing further these pros/cons would be valuable.

Additional minor comments:

Line 124: hotspots instead of "hotpots"?

Line 125: ",)"?

Line 144: "tagO", italic?

Line 236: "S. aureus", italic?

Line 326: "XX", 738 + strains from the study?

Version 1:

Reviewer comments:

Reviewer #2

(Remarks to the Author)

The authors have addressed my concerns.

Reviewer #3

(Remarks to the Author)

Giuleri et al. have addressed all my concerns and added discussion/details to strengthen the arguments and the proposal of the study.

A multi-hospital, clinician-initiated bacterial genomics program to investigate treatment failure in severe *Staphylococcus aureus* infections

Revision of Nature Communications manuscript NCOMMS-24-71326

Point by point response to Reviewer comments

Reviewer #1 (Remarks to the Author):

Giulieri and colleagues report on the utilization of whole genome sequencing strategy to detect treatment failure in patients suffering severe Staphylococcus aureus infection. The authors used comparative genomics techniques to detect emergence/selection of mutations triggering antibiotic resistance in order to adapt optimally antimicrobial treatment. Monitoring the optimal treatment in real-time is challenging but will represent an inestimable benefit for the patient.

General remarks: The paper is short, clear and the results are definitely informative. The field of research is particularly relevant and the work conducted by a renowned group, very active in the field. The work appears rigorous and the manuscript is well-structured. The conclusions are in accordance with the results obtained by the authors. In conclusion, I see no mandatory changes to require and I would only suggest minor changes or comments.

- 1. The manuscript contains a couple of typos (check l. 125; S. aureus not italicized l. 236; always l. 161).*

Response: We have corrected the typos.

- 2. The study requires real-time sequencing; I would suggest the authors to introduce a (short) section about cost of the analysis.*

Response: We thank the reviewer for the suggestion. We have added a short paragraph to the discussion where we provide the estimated cost of the analysis in our laboratory (584 Australian dollars for two paired isolates [US \$370]) and compare it to the mean cost of whole-genome sequencing in a recent systematic review (US \$194) (Lines 250-258).

- 3. The idea of contacting a large panel of ID specialists is really relevant*
- 4. My main comment relies on the information obtained by WGS. I would suggest to add a paragraph about the limitations of the study. Indeed, some resistances are not linked to emergence/selection of mutations, phenotypical tests appear mandatory. In addition, it appears obvious that the method is mainly adapted to persistent infections but not to relapsing ones; in the latter situation the strain could evolved silently for months or years without the possibility to detect any infectious episode.*

Response: We agree with the limitation mentioned by the reviewer, namely that parallel phenotypic testing is necessary to avoid missing emergence of resistance that is not predicted by the detection of de novo mutations. For this reason, our approach includes systematic paired antibiotic susceptibility testing (see methods line 307-315). This will be the case at least until large databases of phenotype-genotype association are available for *S. aureus* mutational resistance. We also discuss this key limitation (lines 245-248). Regarding the second limitation, we agree that evolving strains can be missed when the information is missed, however, based on our methods, they should be available when a clinically significant recurrence is detected. We believe that our approach is particularly suited for recurrences as it allows to: 1) reliably distinguish between relapse with the same strains and reinfection with a genetically distant strain; 2) detect adaptive mutations in the relapsing strain.

5. *I hope a larger study allowing assessing benefit of the method will be initiated in the future by the authors*

Response: We share the reviewer's aspiration to investigate our approach in a larger study. We are in the process of planning a prospective assessment where a standardized genomic report will be provided to clinicians managing antibiotic treatment failure. We plan to quantify the impact of the report on the management decisions related to the case (i.e. choice of salvage treatment) but also to measure turn-around times and to perform a health economic analysis. This vision is now included in the discussion at lines 264-275.

Reviewer #2 (Remarks to the Author):

I appreciate that this is a pilot study of 11 cases, and as such, the data are limited. I applaud the authors for clearly and accurately describing the limitations of this study and the requisite next steps (critically, looking at patient outcomes). While I agree that the technology shows promise, there are several ways in which that case is not strongly made by the manuscript as it is currently presented.

Major:

1) The statement that "Overall, the survey suggests that supplementary phenotypic and genomic information provides added value to the assessment of antibiotic failure and could inform decisions on antibiotic therapy." does not seem strongly supported. The authors must present statistical analysis to indicate whether any meaningful shift in the metrics they are presenting occurred with the presentation of users with genomic data for any of the metrics they are collecting (ie the data in Figure 3, panels B through F).

If these shifts in provider decision making do not pass statistical significance in this small pilot study it is not necessarily a deal-breaker, but this statistical analysis should be performed and presented. If statistical significance is not achieved, are there features of particular cases where genomic data was most informative?

Response: We thank the reviewer for this important comment. For the 3 main recommendation outcomes (change of antibiotic therapy, source control recommendation and antibiotic duration) we performed a univariate and multivariate regression analysis using following predictor variables: presence of adaptive mutations, genomic report, country of residence and professional profile of the respondents (e.g. clinical microbiology training, bacterial genomics expertise, *S. aureus* expertise). The analysis shows that there was a significant change in the proportion of clinicians recommending antibiotic change after reading the genomic report. The association remained significant after correcting for presence of adaptive mutations and for the profile characteristics of the clinicians. No significant association with source control and antibiotic duration recommendation was found. The results of the statistical analysis are presented at lines 195-201 in the manuscript and in the supplementary file. The statistical methods are described at lines 409-417.

2) Some of the language used is inconsistent with is a bit confusing and needs to be better clarified or defined, in my opinion. Infections are described, alternately, as "persistent" vs "recurrent" vs "re-infection" vs "relapse" vs "adaptive". My understanding (albeit limited) is that "persistence" refers to a state where a small subpopulation of bacteria, called "persisters," remain alive and dormant within the body even after treatment, potentially leading to a relapse of the infection later on, while "recurrence" more generally means the infection comes back again, which could be due to either persistence (same bacteria reactivating from a dormant state) or reinfection with a new strain of the bacteria from an external source. In this framework, recurrence and persistence are not mutually exclusive. This is all a way of saying that the authors should formally define persistence and recurrence in the abstract, and relate these definitions to both the empiric measurements they are making about total and adaptive mutations, and to the other terminology they are using (ie, relapse).

Response: In this study we use a clinical definition of persistent or recurrent infection, i.e. microbiologically documented treatment failure, or positive follow-up cultures before treatment response (persistent infection) or after initial treatment response (recurrent infection). We agree with the reviewer that this can create confusion with the biologically defined concept of persisters. To

avoid this confusion, we have replaced the word “persistence” with “persistent infection” throughout the manuscript.

In addition to the first clinically defined layer, we use genomics to separate persistent/recurrent infections in those that are caused by the same strain (i.e. genetically related and inferred to be monophyletic) or by a genetically distinct strain (different lineages or sequence type, non-monophyletic). Using this definition, we separate persistent infection in persistent infection *sensu stricto* and superinfection and recurrent infection in relapse and reinfection.

We have created a supplementary figure with these important definitions (Figure S1). In addition, these definitions are provided in figure 1, abstract (lines 44-46) and introduction (lines 65-68) or first section of results (lines 108-11) + methods (lines 302-303 and line 361).

3) *Line 154: "Overall probability of adaptive mutations was 30 (interquartile range 20-70). " This statement is unclear to me, and seems to conflict with the following statement that "adaptive mutations were correctly predicted only for two out of four episodes ".*

Response: When assessing the clinicians' estimation of the probability of adaptive mutations, we first considered summary metrics across all 11 cases: median estimated probability was 30%, with an interquartile range 20%-70%. This value indicates that there is a wide range of estimates across clinicians but we agree with the reviewer that it is not very informative in itself. Therefore, we looked at cases with adaptive mutations and asked ourselves the question of how good clinicians can predict them. We reasoned that if clinicians could predict adaptive mutations with high accuracy, our analysis would be not very useful. To assess this, we considered that adaptive mutations were predicted correctly if more than 50% of the clinicians indicated an estimated probability above 50%, that is if the median probability for the episode was above 50%. We discovered that only two out of four episodes has a median estimated probability above 50%, indicating that it is not possible to predict adaptive mutations without genomic information. We clarify this analysis and our rationale to perform at lines 186-193.

4) *Figure 3. "Clinicians assessment of probability of adaptive mutations" is being measured. It is unclear why this metric is useful or is being tracked. Shouldn't the empiric data define whether there are adaptive mutations present in the strains being examined? Why and how is this the opinion of the provider receiving the report, instead of a presentation of fact?*

Response: See our response above.

5) *In the case of emerging antibiotic resistance, adaptive mutation would be of obvious importance, but it is not clear how S. Aureus adaptations during chronic disease that modulate immune response pressure (eg, agrA) are useful. As it relates to informing clinical decision making, I feel that the authors need to more cleanly focus on adaptive mutations relating to antibiotic resistance rather than those related to other stigmata of chronic disease, or at least to divide such mutations into discrete categories.*

Response: We agree with the reviewer that it is important to distinguish mutations in genes associated with antibiotic resistance from other mutations that are primarily linked to pathogenesis or metabolism. Indeed, while agr mutations are strongly associated with adaptation, their clinical impact remains controversial. For this reason, we have classified mutations in Figure 2 according to the main functional association of their gene. The functional classification was based on gene annotations in Microbesonline and review of the literature (Source Data file, Methods lines 367-373)

6) *The form of the analysis report is described, but with only 11 cases contained in this study it would be beneficial and appropriate for the authors to present the actual reports issued for each of the 11 cases (ie, the pair of reports issued to the clinicians) as supplemental information.*

Response: We have provided reports with the supplementary information (Supplementary Table S2).

7) *Genomic data availability: The sequencing data generated from this study should be made publicly*

available in the SRA or equivalent database.

Response: We have uploaded the sequence reads of the 60 isolates included in the study into the NCBI sequence reads archive SRA under the project accession number PRJEB85744. This information is provided in the Data availability statement and in the Source Data file.

Minor:

1) *Figure 1 legend is incomplete "Maximum likelihood phylogenetic tree of XX S. aureus bacteraemia strains".*

Response: We have completed the legend of Figure 1.

2) *Line 264. The source of the "closest external isolate" used is unclear. Is this a genome from a public repository unrelated to this study, or was some operation performed to identify an isolate from the study cohort that satisfied this requirement? This is alluded to in the figure legend but should be cleanly communicated in the methods.*

Response: For each episode we used the global phylogeny (all genomes from this study + 738 *S. aureus* bacteremia genomes that were previously published by our group) to infer the most recent common ancestor (MRCA) of the episode. We then extracted the parent node of the MRCA, obtained its descendant tips and selected the one genome that didn't belong to the episode of interest. We have added an explanation at lines 379-383.

3) *line 176 : "Our survey showed the added value of additional phenotypic and genomic investigations and suggested that the awareness of these results can inform treatment decision in up to 30% of cases. " How is that statistic/conclusion being reached? Does this refer to the 34% of responses in changes in antibiotic choice?*

Response: Yes, this number refers to the changes in antibiotic choices. To further clarify, the source data for figure 3 have been added.

Reviewer #3 (Remarks to the Author):

The study by Giuleri et al. aims to show the usefulness and potential of clinical microbial (bacterial) genomics in real-time to identify and track bacterial adaptive mutations related to antibiotic treatment failure. The identification of pathoadaptive mutations driving oxacillin resistance as well as the discrimination between persistence/relapse/reinfection using WGS has been shown. Furthermore a roadmap for its use is proposed based on the interaction with clinicians.

The study is methodologically correct, well structured and aims to support/encourage the implementation of microbial genomics in the clinic aiming for a genomics-informed treatment and not only a posteriori analysis. Though most of my questions have been answered in the manuscript, some key aspects might require further discussion/detail to strengthen the arguments and the proposal. Below I provide some comments/questions that when addressed might help refine the manuscript.

-As one of the factors preventing the implementation of bacterial genomics within the clinical practice is mentioned "turn-around-times that are not suitable for acute infections", however no data is provided regarding a timeline (crucially duration) from sample acquisition to genomic analysis results being provided.

Response: We agree this is an important point. However, we haven't collected data on the turn-around time (TAT) between referral of the isolates and genomic report for this study. In the discussion we provide an estimate for our laboratory and discuss the issue of turn-around time more broadly, as TATs are very location and laboratory specific, and currently highly variable (lines 258-264).

-Considering the increased accuracy provided by the novel ONT flow cells, the added value of having long reads to identify genetic rearrangements and the known low turn-around-time. Do the authors

believe it would be possible to use this alternative technology to perform the same analysis? I believe adding this to the discussion would be valuable. Specially interesting for those having this technology already.

Response: We agree that the option of using ONT sequencing is an important question. We have added a paragraph to the discussion (lines 277-288). While we acknowledge the potential advantage of ONT, we believe that at this stage short-read sequencing is more appropriate for within-host evolution analysis. More studies comparing the performance and cost of short and long-read sequencing in this setting should be performed

-An additional column in table 2 specifying Month/Year of the episode would help to easily follow the data in the figures.

Response: We have added a column with month/year as suggested. The information is also available in Source Data file.

-Considering that all samples do not belong to the same CC. Why was chosen a SNP analysis over a cgMLST analysis for the ML phylogenetic tree of Figure 1B?

Response: We agree that our global phylogeny is limited by the diversity of clones represented. However, we believe that our approach is still valid for the purpose of this study, which was to confirm that same-patient isolates were mono-phyletic and therefore indicative of persistent infection. Although we recognised that per-CC phylogenies are more accurate, using a SNP-based core genome alignment with a single reference is consistent with published work on *S. aureus* (see for example: Aanensen. mBio 2016, DOI: <https://journals.asm.org/doi/10.1128/mbio.00444-16> or Young, Microb Genom 2021, DOI: <https://doi.org/10.1099/mgen.0.000700>). In addition, the approach used here uses a soft definition of core genome (sites with up to 10% gaps) that is expected to increase the size of the alignment and the resolution of the analysis.

We were interested in assessing the value of a cgMLST approach in inferring persistent infections. We performed a cgMLST analysis of the genomes included in the global phylogeny and found that the method performed similarly to a pairwise SNP distance analysis, as previously observed by Lagos et al (DOI: <https://doi.org/10.1038/s41598-022-14640-w>). The results of this analysis are presented at lines 139-148, in figure S2 and the approach is described at lines 342-352.

-Considering that there are cases where not additional information was provided by WGS analysis (i.e. phenotype changed and association with genomic data not possible, such as the case in hospital A, Mar 2023) and that the costs of performing WGS are still considerable. Is the data presented enough for a strong argument supporting the implementation of WGS to support treatment decisions? Discussing further these pros/cons would be valuable.

Response: We thank the reviewer for this comment and agree that the cost-effectiveness of WGS requires further study. We discuss this at lines 271-275 and propose avenues to increase cost-effectiveness, including a targeted approach based on clinical criteria, explore the use of Maldi-TOF to infer genetic relatedness. Re case A/Mar 2023: the phenotypic change was most likely a lab artifact, as it is known that penicillin susceptibility testing can be unreliable. Here the key analysis was the lack of detection of the *blaZ* gene, highlighting how WGS can provide information even when no evolution/adaptation is detected.

Additional minor comments:

Line 124: hotspots instead of "hotpots"?

Line 125: ",)"?

Line 144: "tagO", italic?

Line 236: "S. aureus", italic?

Line 326: "XX", 738 + strains from the study?

Response: We have edited the manuscript to address the comments of the reviewer.